# Ceftazidime/Avibactam in Ventilator-Associated Pneumonia Due to Difficult-to-Treat Non-Fermenter Gram-Negative Bacteria in COVID-19 Patients: A Case Series and Review of the Literature

**DOI:** 10.3390/antibiotics11081007

**Published:** 2022-07-26

**Authors:** Giulia Jole Burastero, Gabriella Orlando, Antonella Santoro, Marianna Menozzi, Erica Franceschini, Andrea Bedini, Adriana Cervo, Matteo Faltoni, Erica Bacca, Emanuela Biagioni, Irene Coloretti, Gabriele Melegari, Jessica Maccieri, Stefano Busani, Elisabetta Bertellini, Massimo Girardis, Giulia Ferrarini, Laura Rofrano, Mario Sarti, Cristina Mussini, Marianna Meschiari

**Affiliations:** 1Department of Infectious Diseases, Azienda Ospedaliero-Universitaria of Modena, 41124 Modena, Italy; g.burastero@gmail.com (G.J.B.); gabriella.orlando7@virgilio.it (G.O.); antonella.santoro7@gmail.com (A.S.); marymenozzi@gmail.com (M.M.); ericafranceschini0901@gmail.com (E.F.); andreabedini@yahoo.com (A.B.); adriana.cervo@gmail.com (A.C.); matteo.faltoni@gmail.com (M.F.); erica.bacca@gmail.com (E.B.); 2Department of Anaesthesia and Intensive Care, Azienda Ospedaliero-Universitaria of Modena, 41124 Modena, Italy; emanuela.biagioni@gmail.com (E.B.); irenecoloretti@gmail.com (I.C.); melegari.gabriele@aou.mo.it (G.M.); maccieri.jessica@gmail.com (J.M.); stefano.busani@unimore.it (S.B.); e.bertellini@ausl.mo.it (E.B.); massimo.girardis@unimore.it (M.G.); 3Clinical Microbiology Unit, AUSL, Modena, Via Giardini 1355, 41126 Baggiovara, Italy; giulia.ferrarini84@gmail.com (G.F.); laura.rofrano@libero.it (L.R.); 4Clinical Microbiology Laboratory, Azienda Ospedaliero-Universitaria of Modena, 41124 Modena, Italy; sarti.mario@aou.mo.it; 5Clinic of Infectious Diseases, Department of Infectious Diseases, University of Modena, 41124 Modena, Italy; cristina.mussini@unimore.it

**Keywords:** non-fermenter gram negative, VAP, ceftazidime/avibactam

## Abstract

Ventilator-associated pneumonia (VAP) in critically ill patients with COVID-19 represents a very huge global threat due to a higher incidence rate compared to non-COVID-19 patients and almost 50% of the 30-day mortality rate. *Pseudomonas aeruginosa* was the first pathogen involved but uncommon non-fermenter gram-negative organisms such as *Burkholderia cepacea* and *Stenotrophomonas maltophilia* have emerged as other potential etiological causes. Against carbapenem-resistant gram-negative microorganisms, Ceftazidime/avibactam (CZA) is considered a first-line option, even more so in case of a ceftolozane/tazobactam resistance or shortage. The aim of this report was to describe our experience with CZA in a case series of COVID-19 patients hospitalized in the ICU with VAP due to difficult-to-treat (DTT) *P. aeruginosa*, *Burkholderia cepacea*, and *Stenotrophomonas maltophilia* and to compare it with data published in the literature. A total of 23 patients were treated from February 2020 to March 2022: 19/23 (82%) VAPs were caused by *Pseudomonas* spp. (16/19 DTT), 2 by *Burkholderia cepacea*, and 6 by *Stenotrophomonas maltophilia*; 12/23 (52.1%) were polymicrobial. Septic shock was diagnosed in 65.2% of the patients and VAP occurred after a median of 29 days from ICU admission. CZA was prescribed as a combination regimen in 86% of the cases, with either fosfomycin or inhaled amikacin or cotrimoxazole. Microbiological eradication was achieved in 52.3% of the cases and the 30-day overall mortality rate was 14/23 (60.8%). Despite the high mortality of critically ill COVID-19 patients, CZA, especially in combination therapy, could represent a valid treatment option for VAP due to DTT non-fermenter gram-negative bacteria, including uncommon pathogens such as *Burkholderia cepacea* and *Stenotrophomonas maltophilia*.

## 1. Introduction

A significant incidence of ventilator-associated pneumonia (VAP) has been reported in SARS-CoV2 patients admitted to the Intensive Care Unit (ICU) during the COVID-19 pandemic [1]. Some authors reported VAP incidence peaking at 40% with an incidence density of 28/1000 ventilator days [2,3,4].

Factors such as long hospitalization, prolonged mechanical ventilation, and immunosuppression contributed to this overall increased incidence of VAP [5]. Compared to gram-positive bacteria, gram-negative bacilli have been found responsible for the majority of VAP (19.5% vs. 83.6%) with *Pseudomonas aeruginosa* (22.3%), *Enterobacter* spp. (18.8%), and *Klebsiella* spp. (11.5%) among the most commonly identified pathogens [3].

Moreover, in this specific scenario of VAP superinfection in critically ill COVID-19 patients, several studies reported an unusually high incidence of *Burkholderia cepacea* and *Stenotrophomonas maltophilia* among gram-negative pathogens [2]. The prevalence of VAP due to multi-drug resistant (MDR) isolates varies from 23 to 35% with large differences among countries [3,6].

Indeed, according to the European Surveillance of Antimicrobial Resistance in Europe, the percentage of carbapenem-resistant *P. aeruginosa* in 2020 was below 5% in four countries (10%), whereas six countries (15%) reported percentages equal to or above 50% [7].

The 2016 Infectious Diseases Society of America (IDSA) guidelines on the empirical management of VAP recommended starting with at least two agents active against gram-negative organisms, including *P. aeruginosa* [8]. The advent of two novel β-lactam–β-lactamase inhibitor combinations (BLBLICs), namely, ceftazidime/avibactam (CZA) and ceftolozane/tazobactam (C/T), has broadened the treatment options for patients with suspected MDR organisms [5].

Based on controlled clinical trials, both these drugs were approved for the treatment of VAP caused by *P. aeruginosa* and other *Enterobacteriaceae* [9,10], but unfortunately, a dramatic worldwide shortage occurred in December 2020 when C/T was withdrawn following bacterial contamination that occurred during the manufacturing process [11]. In this scenario, CZA became one of the best remaining options for treating infections provoked by carbapenem-resistant microorganisms.

Unfortunately, data on the efficacy of CZA in bacterial infections caused by Non-Fermenter Gram Negative (NFGN), are limited. Therefore, our aim was to describe our real-life experience with CZA, both alone and in combination regimes, among critically ill COVID-19 patients with VAP due to DTT *P. aeruginosa*, *Burkholderia cepacea*, and *Stenotrophomonas maltophilia*, and to review the current published literature on this emergent issue.

## 2. Materials and Methods

### 2.1. Design and Ethics Approval

We conducted a retrospective, observational clinical case series including 23 COVID-19 patients with VAP caused by carbapenem-resistant NFGN bacteria (*Pseudomonas* spp., *Stenotrophomonas maltophilia*, and *Burkholderia cepacea*) and admitted to two different ICUs in the University Hospital of Modena from February 2020 to March 2022. All the patients included in our series were treated with CZA alone or in combination therapy.

For each patient, we described the demographics and clinical characteristics of VAP, including the severity of disease, etiology, type, and duration of empirical and target therapies. Microbiological cure, relapse, and 14/30-day mortality, other than overall mortality, from the ICU admission to the end of the follow-up were assessed. Moreover, to compare our data with those published in the literature, a review of the recent literature was performed. 

Our local Institutional Review Board (IRB) approved the present clinical report with the following protocol number 484/2021/OSS/AOUMO SIRER ID 2556. Given the descriptive nature of the paper, informed consent has been waived by the IRB. Data were collected for the purpose of health care according to the standard treatment procedure.

### 2.2. Definitions

*Pseudomonas* spp. strains were categorized as difficult-to-treat (DTT) according to the last definition proposed by IDSA guidelines [12]. Indeed, in order to optimize the phenotypic definition of MDR pathogens, a new definition of difficult-to-treat (DTT) bacteria has been recently proposed considering a pathogen being intermediate or resistant to all reported agents in carbapenem, β-lactam, and fluoroquinolone categories [13]. In particular, for *P. aeruginosa*, a DTT resistance was defined as non-susceptibility to all of the following: piperacillin-tazobactam, ceftazidime, cefepime, aztreonam, meropenem, imipenem-cilastatin, ciprofloxacin, and levofloxacin [12].

Considering how challenging it is to make an accurate diagnosis of VAP in COVID-19 patients with preexisting lung injuries due to the viral disease, VAP was defined using a modification of the European Centre for Disease Control definitions as a combination of radiological, clinical, and microbiological criteria in a patient who has been receiving mechanical ventilation for at least 48 h [2,14].

The microbiological cure was defined as the absence of the same gram-negative bacilli (GNB) isolates, both assessed within 7 days and at the end of treatment (EOT) with CZA. We also evaluated, in patients who previously reached clinical and microbiological cure, the occurrence and the time onset of relapse of the clinical signs and/or symptoms or the microbiological recurrence of the baseline pathogen from an appropriate specimen. Mortality was described within 14 and 30 days from the start of treatment. Overall mortality evaluated until March 2022 was also recorded for each patient.

CZA was administered at a standard dose of 2.5 g every 8 h in a 3-h infusion or 7.5 g in a 24 h continuous infusion, diluted in 100 mL of saline solution, with a renal adjustment dose according to the SPC of the medicinal product. The choice to use single or combination therapy was made by an infectious disease specialist during antimicrobial stewardship interventions based on patients’ clinical conditions, etiology (mono or polymicrobial infection), and microbiological characteristics, including sensitivity (CZA MIC) of the isolates collected.

### 2.3. Microbiological Methods

All collected isolates were identified by MALDI-TOF MS using VITEK MS (bioMérieux, Marcy l’Etoile, France) following the manufacturer’s instructions. Antimicrobial susceptibility testing was performed by VITEK MS (bioMérieux, Marcy l´Etoile, France) and CZA was confirmed by broth microdilution panel YEUMDROF (Thermo Fisher DiagnosticsS.p.A., Rodano, Italy). MICs were interpreted according to the EUCAST breakpoints, Version 11.0, 2021.

### 2.4. Statistical Analysis

Descriptive statistics were performed; continuous variables were presented as number (n), median and interquartile range (IQR), minimum (min), and maximum (max), while categorical variables were presented as frequency/percentage (n/%). A Kaplan–Meier curve was performed concerning 30-day overall survival analysis, starting from the day when Ceftazidime/avibactam was started for treatment. Statistical analysis was performed using STATA^®®^ software version 14 (StataCorp. 2015. Stata Statistical Software: Release 14. College Station, TX, USA: StataCorp LP).

## 3. Results

A total of 23 patients were included; the median age was 69 years old (IQR 64–76.5), 30% were female, and all had a BMI > 24.99 kg/m^2^ (IQR 27–32).

Patients’ characteristics and clinical features are shown in Table 1. The median duration of ventilation was 47 days (IQR 35.5 −58.5), and VAP occurred after a median of 29 days (IQR 19–40) from ICU admission and after 22 days from endotracheal intubation, respectively (IQR 15.5–28), septic shock was the clinical presentation in more than half of the patients (15/23, 65.2%) with a median SOFA score of 8 (IQR 7–9.5). In total, 7/23 (30%) patients received continuous renal replacement therapy (CRRT) and 2 patients received Extra Corporeal Membrane Oxygenation (ECMO).

Concerning pathogens, *Pseudomonas* spp. was isolated in 19/23 (82.6%) of the samples (17 *P. aeruginosa*, 1 *P. putida,* and 1 *P. fluorescens**)* with a DTT profile in 16/19 (84.2%) of the cases and the 3 remaining cases had elevated meropenem MICs as well as polymicrobial infections. 

*Stenotrophomonas maltophilia* was found in 6/23 (26.0%) and was resistant to trimethoprim/sulfamethoxazole in 50%.

*Burkholderia cepacia* was isolated in 2/23 (8.6%) patients; of which, 1 case showed a DTT profile and the other had coinfection with an ESBL-producing *E. cloacae*.

Finally, in 12 out of 23 (52.1%) VAP cases, the bronchoalveolar specimen collected from the low respiratory tract showed a polymicrobial infection (in 3/23 VAP, both *Pseudomonas* spp. and *S. maltophila* were isolated). 

Importantly, 11/23 (47.8%) patients showed rectal colonization with DTT *P. aeruginosa*, 2/23 with ESBL and KPC *K. pneumoniae*, respectively, and 1 with ESBL *E. cloacae*.

Microbiological isolates are shown in Table 2.

In our case series, a high dose of dexamethasone was administrated in all patients as standard of care for SARS-CoV2 pneumonia and 17/23 (73.9%) received an antibody against IL-6 receptor (Tocilizumab).

CZA was administrated in 2/23 patients with intermittent infusion (II) over 2 h of 2.5 g; in all the others, the administration was performed by extended infusion (EI), 5 g every 12 h.

The median duration of infusion was 9 (IQR 6.5–12) days and in 21/23 (91.3 %) patients CZA was used in combination therapy; the most frequent association was with fosfomycin in 9 patients, with meropenem and aminoglycosides (often with inhaled amikacin) in 5 and 3 patients, respectively, with both meropenem and fosfomycin in 1 patient. Finally, trimethoprim-sulfamethoxazole was associated in 2 cases and aztreonam in another.

Among patients in ECMO and CVVH, 1.25 g every 8 h of CZA were infused.

Concerning outcomes (reported in Table 3), a microbiological cure was achieved in 11/21 (52.3%) (data were not available for 2 patients) Notably, 4/11 experienced a microbiological recurrence with clinical relapse and isolation of CZA-resistant *P. aeruginosa* in 1 case.

As shown in Figure 1, the 30-day overall mortality rate was almost 65%, while the 14-day mortality rate was almost 35%. The higher 30-day mortality rate was 61.1% for VAPs due to *Pseudomonas* spp., followed by 50.0% and 16% for *Burkholderia cepacia* and *Stenotrophomonas maltophilia*, respectively.

## 4. Discussion

In the recent scenario dominated by the COVID-19 pandemic, VAP represented the most fatal bacterial complication among critically ill patients, requiring accurate management and appropriate therapy.

Our case series highlights the extreme complexity of critical COVID-19 patients. Among these patients, recurrent superinfections resulting from the long incubation period were characterized by the selection of difficult-to-treat and MDR organisms as well as unusual pathogens such as the *Burkholderia cepacia*, only described before in patients with cystic fibrosis, and *Stenotrohomonas maltophilia* typically isolated in the hematological setting.

The high 30-day mortality rate reported in our cases (60.8%) deserves important consideration. Six patients were in CVVH during CZA infusion and this variable was demonstrated to be associated with higher clinical failure [15]. Moreover, more than half of the cases included in our series developed VAP during the first wave of the COVID-19 pandemic, when SARS-CoV2 pneumonia and its respiratory features were not characterized [16]. It is important to note that the antivirals or monoclonal antibodies now routinely used to prevent worse clinical evolution in COVID-19 patients were not available during the first wave. Finally, in five patients, death occurred long after 30 days from the end of therapy: this datum seems to exclude a direct role of therapy and supports our hypothesis that is essential to consider the role of underling COVID-19 pneumonia in the evaluation of overall mortality.

The substantial increases in VAP incidence together with the worrisome rate of mortality, exceedingly even 50% in many studies [17], could be related to several factors. First, patients with COVID-19 admitted to ICU are generally severely hypoxemic, with both parenchymal and microvascular lung damage [6]. Secondly, patients with COVID-19 frequently needed prolonged mechanical ventilation, prone positioning [18], and received immunomodulant therapies. Moreover, due to intensive workload and increments of beds, a large part of the ICU staff was reallocated, and newly recruited healthcare workers had inadequate training in the prevention of cross-contamination leading to lower adherence to infection control standards and VAP prevention bundles [19]. Finally, regarding pathogen-related risk factors, it is well known that recurrent infections with *Pseudomonas aeruginosa* and/or *Burkholderia cepacia* could accelerate the functional pulmonary decline that increased morbidity and mortality [20,21].

Although data exist supporting the use of CZA in these particularly challenging pulmonary infections, the in vitro and clinical efficacy of CZA, alone or combination regimens as rescue therapy, was understudied.

According to the International Network for Optimal Resistance Monitoring (INFORM) global surveillance program [9,22,23,24], CZA demonstrated potent in vitro activity against extended-spectrum β-lactamases (ESBLs), AmpC β-lactamases, KPC, and OXA-48-producing *Enterobacterales* other than against metallo-β-lactamase (MBL)-negative P. aeruginosa. Indeed, the in vitro activity of CZA against MBL-producing pathogens was very limited (MIC90, 128 lg/mL) [24].

Regimes based on the use of CZA have been already demonstrated to be more effective than other available antibiotic agents for the treatment of infection caused by class A carbapenemase (KPC) producing *K. pneumoniae* [25] in critically ill mechanically ventilated patients [26]. However, few data are available concerning its efficacy in the treatment of DTT non-fermenter gram-negative (NFGN) bacteria such as *Pseudomonas* spp., *Burkholderia cepacia*, and *Stenotrophomonas maltophilia* which are frequently associated with VAP [27,28].

The treatment of infections due to DTT NFGN represents a particularly difficult challenge for the intrinsic pattern of antibiotic resistance and the very few available antibiotic options. While the activity of CZA against MDR *P. aeruginosa* is reported in vitro and animal models [9,10], no data about clinical efficacy are available from randomized controlled trials. In such a complicated scenario, data from real-life experience may play an important role, and indeed, encouraging data was published regarding the treatment of infections due to MDR gram-negative, with particular regard to *P. aeruginosa* and *Burkholderia cepacea* [20,29,30].

In studies presented by Dimelow et al. and Nicolau et al. [31,32], the concentration of CZA in the epithelial lining fluid (ELF) is approximately 30% of the plasma concentration, resulting in adequate clinical efficacy. However, the dose of 2.5 g every 8 h as a 2-h infusion resulted in 100% effectiveness on sensitive *Pseudomonas* isolates [33], while the same dosage may not be sufficient to achieve adequate concentrations in the lungs of patients affected by DTT *Pseudomonas aeruginosa* with suboptimal MICs. Indeed, among our cases, four isolates of *Pseudomonas* tested had CZA MIC values > 8 mg/L and the other four strains had a CZA MIC of 8 mg/L, a value close to EUCAST clinical susceptibility breakpoints [34]. This could be affected the clinical outcomes, however, the more appropriate selection of CZA dosage regimen (combination therapies and EI) may have contributed to overcoming these in vitro limits. The encouraging PK/PD data related to the negative association between EI and mortality seem to suggest that this method of infusion could not only maximize clinical efficacy but also prevent the occurrence of resistance development [35,36]. 

Another widely debated issue on CZA treatment in critically ill patients concerns the need for dosage adjustment according to renal function [37]. Indeed, phase III clinical trial patients with moderate renal impairment and deep infections experienced a decrease in drug efficacy, potentially as a result of rapidly improving renal function during therapy with consequent CZA underdosing [38]. A worse outcome was assessed among patients affected by septic shock, especially if pulmonary, and in continuous renal replacement therapy (CRRT) with CZA [15]. In this scenario, a practical review was recently published with the aim to guide dose optimization of novel antibiotics, such as CZA, for the management of multidrug-resistant gram-positive and gram-negative infections during CRRT in critically ill patients, and an increased dosage of CZA in this setting is proposed to achieve positive clinical outcome [15].

In the following paragraphs, we review published evidence supporting the use of CZA in ICU patients with VAP sustained by DTT NFGN, focusing on DTT-*Pseudomonas* spp. *Burkholderia cepacia*, and *Stenotrohomonas maltophilia*.

### 4.1. Ceftazidime-Avibactam for the Treatment of DTT Pseudomonas aeruginosa Pulmonary Infections among Critically Ill COVID-19 Patients

*P. aeruginosa* is a non-fermenting gram-negative responsible for 4 to 14% of healthcare-associated infections and 16 to 40% of cases of VAP [39]. The recent literature describing bacterial co-infections in patients hospitalized with COVID-19 showed *P. aeruginosa* to be among the most frequently identified species, with a higher proportion in critically ill ICU patients [40]. *P. aeruginosa* can express numerous acquired antimicrobial resistance mechanisms, virulence factors, and mechanisms for evading host defenses. The recent data published by European Antimicrobial Resistance Surveillance Network showed that 31.8% of *P. aeruginosa* isolate strains were resistant to at least one of the first live antimicrobial classes with a potential anti-pseudomonas activity, while MDR and an XDR phenotype with resistance to two or more antimicrobial classes were found in 17.6% of isolates and 3.4% of the isolates, respectively. In this scenario, CZA represents a valuable weapon as evidenced by both several surveillance [9,23,41,42,43] and clinical studies [26,44,45,46,47].

Sader et al. analyzed the susceptibility of CZA against gram-negative bacteria from ICU and non-ICU patients, including those with VAP. In this study, CZA inhibited 95.6% and 97.3% of *P. aeruginosa* isolates from ICU patients and VAP, respectively, and 80.7% of ceftazidime-non-susceptible strains. Furthermore, CZA exhibited promising activity against MDR and XDR strains, inhibiting 80.7% and 74.5% of isolates at a MIC of ≤8 mg/L [48]. Nevertheless, the report of the emergence of CZA resistance is progressively increasing after its current clinical use, and this highlights the need for careful monitoring for the development of resistance. In a retrospective study, collecting 111 MDR/XDR *Pseudomonas aeruginosa* isolates in our university hospital, the CZA susceptibility rate was 42.1% [49]. The same results were also confirmed in a more recently published German Multicenter Study [50]. Concerning real-life experiences, although limited, favorable outcomes with CZA treatment have been reported in some patients with MDR and XDR *P. aeruginosa* infections. Daikos et al. performed an updated overview of CZA treatment for *P. aeruginosa* infections, concluding that CZA may have a potentially important role in the management of serious and complicated *P. aeruginosa* infections, including those caused by MDR and XDR strains [51]. However, due to study designs, most retrospective studies are non-comparative and based on small samples so the role of CZA in this setting remains highly debated. The IDSA guidelines recommended treatment of severe infections due to DTT *P. aeruginosa* with ceftolozane-tazobactam, imipenem- relebactam, and ceftazidime-avibactam as monotherapy [12]. In contrast, the new ESCMID guidelines, due to insufficient evidence, do not consider CZA as a possible therapeutic option for treatment of severe VAP/HAP caused by MDR/XDR *P. aeruginosa* [52], while only ceftolozane-tazobactam was suggested if active in vitro [53].

To overcome the above-mentioned limitations of CZA in vivo, such as lung penetration or intermediate susceptibility, although not routinely recommended outside of metallo-β-lactamase producers, a possible option would be to use CZA in combination regimes. Across the literature, combination therapy was associated with lower mortality than monotherapy in high-mortality-risk patients, especially those with septic shock [54]. Nevertheless, the superiority of combination therapy for DTR-CRPA is still controversial. In contrast to very low-certainty evidence for an advantage of combined polymyxin [55], a recent study found that the combination of CZA-fosfomycin was superior to either drug alone in infected patients with high bacterial burdens due to MDR *P. aeruginosa* that lack metallo-β-lactamases [56]. 

To the best of our knowledge, this is the first report on the efficacy of CZA for the treatment of VAP by DTT *P. aeruginosa* in patients with coexisting severe compromised respiratory function due to SARS-CoV2 infection. We found a 30-day mortality of 61.1% in the subgroup of patients with *P. aeruginosa* infection. This rate is in line with a recently published review by Bassetti et al., where mortality reached 75% for patients with VAP due to MDR pathogens [31]. On this basis, we speculate that the relatively high mortality in our case series could depend on coexisting severe lung damage due to SARS-CoV2 in patients with VAP caused by DTT pathogens; CVVH might have worsened the clinical outcome in a proportion of our patients. 

Future research is needed to explore this issue. Studies with a prospective design and proper statistical power are highly needed, recruiting patients with DTT *P. aeruginosa* infections and aiming to characterize the optimal use of CZA. In particular, the dilemma between monotherapy versus combination therapy necessitates dedicated investigations as much as the definition of the optimal dosage needed to reach the clinical cure in the CRRT setting.

### 4.2. Ceftazidime-Avibactam for the Treatment of Stenotrohomonas maltophilia VAPs

*S. maltophilia* is an aerobic, non-glucose fermenting, gram-negative bacillus that is ubiquitous in water environments [57]. Although often considered a colonizing pathogen, due to its ability to produce biofilm and the impressive number of antimicrobial resistance genes and gene mutations it carries [58], its treatment can be challenging, especially in patients with underlying pulmonary conditions. 

The best known risk factors for *S. maltophilia* infection include chronic respiratory diseases, especially cystic fibrosis, hematologic malignancy, chemotherapy-induced neutropenia, organ transplant, human immunodeficiency virus (HIV) infection, hemodialysis, and being a neonate [59]. Nevertheless, this pathogen is increasingly being isolated among critically ill patients as well.

For these reasons, it is not surprising that some authors reported the significant increasing relevance of this pathogen in patients with SARS-CoV2 infection, particularly in those with prolonged mechanical ventilation, with evidence of increasing incidence of VAP [3].

In this population, *S. maltophilia* can be a true pathogen, promoting the development of hemorrhagic pneumonia or bacteremia and can be associated with high morbidity and mortality.

The IDSA guidelines do not provide a recommended antibiotic regimen for *S. maltophilia* infections because there is no evidence of the best available treatment, and data to determine the additive benefit of commonly used combination therapy regimens remain incomplete [60].

Trimethoprim/sulfametoxazole (TMP-SMX) monotherapy is the preferred treatment agent suggested for mild susceptible *S. maltophilia* infections; minocycline, tigecycline, or cefiderocol in monotherapy can also represent a suitable option because there is no clear evidence that these molecules are associated with clinical failure more than TMP-SMX.

In the case of moderate to severe infections, the use of combination therapy is suggested with a second agent added to TMP-SMX, e.g., minocycline, tigecycline, levofloxacin, cefiderocol, or CZA, the latter possibly in combination with aztreonam (AZT) to better contrast the activity of both metallo-β-lactamase L1 and serine β-lactamases-L2 intrinsic to *S. maltophilia.*

The rationale of this recommendation is based on the provided synergism between CZA and AZT in *S. maltophilia* infections, evidenced by a lower level of minimal inhibitory concentration (MIC) when these molecules are tested together [61,62] and from encouraging clinical outcomes obtained in patients with severe pneumonia or bloodstream infection [62,63,64]. Although randomized clinical trials to prove the real effectiveness of CZA in *S. maltophilia* infections are missing, the use of this molecule is considered a reasonable option in a particular clinical setting such as in the hematologic malignancy population as well as in situations where intolerance or resistance to other agents precludes their use.

In our case series, we reported six cases of VAP caused by *S. maltophilia*, resistant to TMP-SMX in half of the isolates, with polymicrobial infection in five patients.

Meropenem, aztreonam, and TMP-SMX were associated in three patients, respectively, while in two patients, a combination with fosfomycin was used; in the other cases, CZA was prescribed in monotherapy.

The 30-day mortality in this specific subgroup was 16% (1/6), significantly lower than data already reported in the literature [65]. This result is quite surprising considering the concomitant SARS-CoV2 pneumonia. One possible reason for this result could be that CZA, best in combination regimes also without AZT, could be a valid and suitable option for the treatment of *S. maltophilia* pneumonia. Indeed, the only patient who died at 7 days of treatment was treated with CZA in monotherapy. A promising CZA association may be with fosfomycin, which has proved safe and effective in our case series and among VAPs due to *S. maltophilia* as well. A recent study by Moriceau et al. seems to support this hypothesis and concludes that CZA for empirical treatments in severe or polymicrobial infections with *S. maltophilia* could be appropriate [66].

Additional studies are needed to confirm this assumption.

### 4.3. Ceftazidime-Avibactam for the Treatment of Pulmonary Infections Caused by Burkholderia cepacia among Critically Ill Patients

*Burkholderia cepacia* is an NFGN bacterium, commonly found in soil or water and known to cause infection in immune-compromised individuals and patients with cystic fibrosis (CF) [67]. Because *B. cepacia* produces a wide variety of potential virulence factors and exhibits innate resistance to many antibiotics, an infection could be associated with an accelerated decline in respiratory function and related increased morbidity and mortality.

In the literature, the interplay between SARS-CoV2 and *B. cepacia* infection is still unclear, but it has been assumed that the severe systemic inflammatory response usually evidenced in patients with “Cepacia syndrome”, could promote a worse clinical evolution in COVID patients, triggering a hyper-inflammatory reaction and causing critical acute respiratory distress syndrome [68].

In line with this assumption, in a recent case series describing an interhospital outbreak of *B. cepacia* VAP, a prolonged time of mechanical ventilation and higher mortality was evidenced in a subgroup of patients with concomitant SARS-CoV2 infection [69].

Regarding the efficacy of CZA for the treatment of *B. cepacia* pneumonia, the most relevant data come from reports describing the clinical experience in adult patients with CF. There are currently no standard treatment recommendations for the intrinsic pattern of antibiotic resistance related to this pathogen and in vitro antibiotic susceptibility is suggested to be tested before starting any treatment [70].

Furthermore, there is a disagreement between the Clinical and Laboratory Standards Institute (CLSI) and the European Committee on Antimicrobial Susceptibility Testing (EUCAST) about the possibility of providing microdilution breakpoints for tested antimicrobial agents (offered just by the CLSI) since there is no clear correlation between minimum inhibitory concentration (MIC) and clinical outcome [71].

Encouraging data come from the literature [71], where TMP-SMX and CZA were the antibiotics with the highest in vitro susceptibility in 64 *B. cepacia* isolates (83% and 78%, respectively) [72].

A positive clinical experience was reported in the case series of Spoletini et al. where antibiotic regimens including CZA appeared to be safe and effective [20]; similarly, a clinical cure was obtained in a case of persistent bacteriemia by *B. cepacia* in a pediatric patient, after a change of antibiotic regimen from meropenem to CZA in continuous-infusion [73]. For these reasons, some authors suggest considering CZA as a standard and suitable option for the treatment of *B. cepacia* infections. In our series, 1 out of 2 patients with *B. cepacia* VAP died within 30 days from the end of treatment: in that case, CZA was infused in combination with fosfomycin, and in association with TMP-SMX in the other. Similar death rates were also reported in the Spoletini et al. case series where 2/5 patients with very poor prognosis died owing to complex underlying lung pathology, despite multiple courses of CZA in combination with other antibiotics. However, the clinical benefits of CZA-based treatment were demonstrated by the reduction and stabilization of infection markers and improved clinical status. Unfortunately, among critically ill COVID-19 patients, immunomodulatory treatment with dexamethasone and tocilizumab considerably reduces the value of biomarkers so their predictive role in defining significant clinical improvement is very limited [74]. 

The high mortality reported in our cases could be explained by the coexisting condition of definitively compromised respiratory function due to SARS-CoV2 infection, more than to *B. cepacia* infection. However, the limited number of cases does not allow definitive conclusions to be drawn. Therefore, in consideration of the promising results obtained by in vitro studies, data are needed to clarify the role of CZA as a suitable and effective option for the treatment of infections mediated by this intrinsic DTT bacterium. 

## 5. Conclusions

The present report could provide useful data from real-life experience in such complex scenarios as VAP in COVID-19 patients regarding the use of CZA in the management of VAP due to non-fermenter gram-negative bacteria. Our case series confirmed the high mortality rate among COVID-19 critically ill patients affected by ventilator-associated pneumonia due to difficult-to-treat non-fermenter gram-negative bacteria. Nevertheless, the severity of the COVID-19 disease and the peculiar pattern of resistance expressed by those pathogens severely limit the available therapeutic options, leading to CZA as the best available rescue treatment.

Our results seem to suggest that optimized PK/PD characteristics, desirable higher doses with extensive infusion and combination regimes, could be the key elements for CZA treatment success in critical patients with VAP infections due to DTT non-fermenting bacteria. We believe that our cases should open the way for future research to help position CZA as an option of choice for non-fermenter gram-negative bacteria without any other available treatments, especially for emerging ones such as *Burkholderia cepacea* and *Stenotrophomonas maltophilia*, which are currently under-investigated.

## Figures and Tables

**Figure 1 antibiotics-11-01007-f001:**
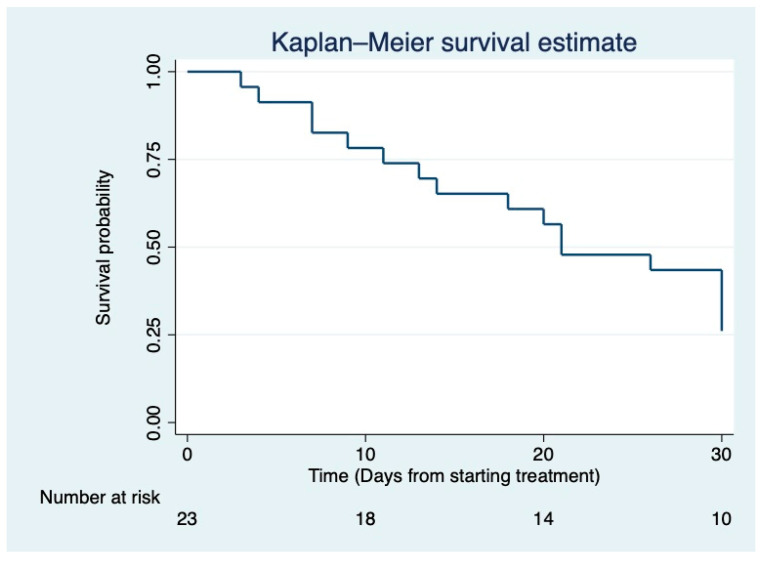
Survival curve showing overall 30-day mortality in COVID-19 patients with ventilator-associated pneumonia due to difficult-to-treat non-fermenter gram-negative bacteria.

**Table 1 antibiotics-11-01007-t001:** Patient characteristics and clinical features.

PT	Age/Gender	ICU Length of Stay before VAP (Days)	Duration of Ventilation before VAP (Days)	SOFA	ECMO/CVVH	Septic Shock
PT1	65/F	29	20	6		no
PT2	72/F	24	24	6	ECMO	no
PT3	52/M	22	17	4	ECMO/CVVH	no
PT4	64/M	17	17	8		yes
PT5	77/F	31	32	8		no
PT6	72/F	57	57	9	CVVH	yes
PT7	72/M	13	8	9		yes
PT8	77/M	36	24	10		yes
PT9	74/M	19	14	8	CVVH	yes
PT10	69/F	29	29	16	CVVH	yes
PT11	77/M	44	27	7		yes
PT12	67/M	15	8	9		yes
PT13	80/M	78	48	6		no
PT14	78/M	7	6	7		yes
PT15	76/M	48	48	9		yes
PT16	68/M	71	12	7		no
PT17	68/M	35	22	7		no
PT18	84/F	95	81	10		yes
PT19	57/M	19	18	10		yes
PT20	40/M	28	26	5		no
PT21	61/M	29	3	7	CVVH	yes
PT22	57/F	26	26	16	CVVH	yes
PT23	64/M	19	18	11	CVVH	yes

PT = patient, ICU = intensive care unit, SOFA = sequential organ failure assessment, CVVH = continuous venovenous hemofiltration, and ECMO = extracorporeal membrane oxygenation.

**Table 2 antibiotics-11-01007-t002:** Microbiological isolates and treatment regimen.

PT	DTT-NFGN/Organism	MIC90 for CZA (mg/L)	Other Organism	Previous/Empirical Treatment Regimen	CZA Regimen	Days of Therapy
PT1	*S. maltophila*	2	*K. pneumoniae ESBL*	FDC then MEM	CZA EI 5 g every 12 h + MEM EI1 g every 8 h	27
PT2	*B. cepacia*	4			CZA EI 5 g every 12 h + FOF 24 g CI	12
PT3	*P. aeruginosa*	2		MEM	CZA EI 1.25 g every 8 h + AMK inhaled	9
PT4	*P. aeruginosa*	2	*S. marcescens*	FEP	CZA EI 5 g every 12 h + FOF 24 g CI	6
PT5	*P. aeruginosa*	8	*K. pneumoniae*	CAZ then TZP then MEM	CZA II over 2 h of 2.5 g + AMK inhaled	18
PT6	*P. aeruginosa*	4	*S. marcescens Colistin-R*	MEM	CZA EI 1.25 g every 8 h + FOF 2 g every 48 h, after a dialytic session	18
PT7	*P. aeruginosa*	16		COL + AMK inhaled	CZA EI 5 g every 12 h + FOF 24 g CI + MEM EI 1 g every 8 h	8
PT8	*P. aeruginosa*	16	*K. pneumoniae KPC*		CZA EI 5 g every 12 h + FOF 24 g CI then FOF was stopped and FDC 2 g EI every 8 h was started	21
PT9	*P. aeruginosa*	16	*K. pneumoniae*		CZA EI 1.25 g every 8 h	3
PT10	*P. fluorescens*	2		MEM	CZA EI 1.25 g every 8 h + MER 1 g EI every 12 h	9
PT11	*P. aeruginosa*	16	*K. pneumoniae ESBL*	MEM + AMK inhaled	CZA EI 5 g every 12 h + FOF 24 g CI, then FOF was stopped and FDC 2 g EI every 8 h was started	5
PT12	*S. maltophila*	16		SXT + AMP	CZA EI 5 g every 12 h + SXT 15 mg/kg/day	11
PT13	*P. aeruginosa*	8		MEM	CZA II over 2 h of 2.5 g + FOF 24 g CI, then FOF was stopped, and FDC 2 g EI every 8 h was started	9
PT14	*P. putida*	8	*S. maltophila*	FEP	CZA EI 5 g every 12 h + FOF 24 g CI	10
PT15	*P. aeruginosa*	4	*S. marcescens ESBL*	MEM	CZA EI 5 g every 12 h + FOF 24 g cCI	9
PT16	*P. aeruginosa*	8	*S. maltophila*	MEM	CZA EI 5 g every 12 h + FOF 24 g CI	25
PT17	*B. cepacia*	16	*E. cloacae ESBL*	MEM	CZA EI 5 g every 12 h + SXT 15 mg/kg/day	9
PT18	*P. aeruginosa*	2	*S. maltophila*	MEM	CZA EI 5 g every 12 h	10
PT19	*P. aeruginosa*	2		TZP	CZA EI 5 g every 12 h + MER EI every 8 h	5
PT20	*S. maltophila*	16			CZA EI 5 g every 12 h + AZT 2 g every 8 h	7
PT21	*P. aeruginosa*	2		MEM	CZA EI 1.25 g every 8 h + AMK inhaled	12
PT22	*P. aeruginosa*	2	*P. aeruginosa*	CAZ then MEM	CZA EI 1.25 g every 8 h + MEM 1 g EI every 8 h	5
PT23	*P. aeruginosa*	2		CAZ then C/T	CZA EI 1.25 g every 8 h + MEM 1 g EI every 8 h	4

EI = extended infusion, II = intermittent infusion, CI = continuous infusion, FDC = cefiderocol, MEM = meropenem, FEP = cefepime, TZP = piperacillin/tazobactam, CAZ = ceftazidime, COL = colistin, AMP = ampicillin, STX = trimethoprim-sulfamethoxazole, CZA = ceftazidime/avibactam, FOF = fosfomycin, and AMK = amikacin.

**Table 3 antibiotics-11-01007-t003:** Patient outcomes.

PT	MC 7	MC EOT	Relapse/ Recurrence (Days after EOT)	Death (Days from the Start of Treatment)	Days until End of Follow Up
**PT1**	no	no		/	359
**PT2**	no	no		26	26
**PT3**	yes	no	yes/7 days	53	53
**PT4**	yes	yes	no	/	459
**PT5**	yes	yes	na	18	18
**PT6**	no	no	no	/	302
**PT7**	no	no		11	11
**PT8**	yes	yes	na	21	21
**PT9**	na	na	na	3	3
**PT10**	no	no		9	9
**PT11**	yes	yes	no	21	21
**PT12**	no	no		37	37
**PT13**	no	no	no	13	13
**PT14**	yes	yes	no	/	411
**PT15**	yes	yes	yes/6 days	/	330
**PT16**	no	yes (*S. maltophila*)	yes/5 days (*P. aeruginosa*)	266	266
* **PT17** *	yes	yes	na	/	11
* **PT18** *	yes	yes	yes/6 days	68	68
* **PT19** *	yes	yes	no	14	14
* **PT20** *	no	no		7	7
**PT21**	no	no		20	20
**PT22**	yes	yes	na	7	7
**PT23**	na	na	na	4	4

MC 7 = microbiological cure within 7 days from start of treatment; MC EOT = microbiological cure at end of treatment, and na = not available.

## Data Availability

The data presented in this study are available on request from the corresponding author.

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
