# Peer review of "Ceftazidime/Avibactam in Ventilator-Associated Pneumonia Due to Difficult-to-Treat Non-Fermenter Gram-Negative Bacteria in COVID-19 Patients: A Case Series and Review of the Literature"

_antibiotics, 2022, doi:10.3390/antibiotics11081007_

Round 1

Reviewer 1 Report

I thank the Editor for giving me the opportunity to review this interesting article of Italian colleagues; substantially, the paper is interesting and deals with an emerging theme, fundamental for intensive care and infectious diseases. Unfortunately, there is a very serious problem that weighs on the whole paper, requiring major revision: basically, it is not clear whether the paper is intended to be a review or a Case Series, as in fact it is.

Title:

- "Case Series" must be entered.

- “Best available treatment” is misleading because a comparison is not performed in the methods. I would rather insert a title like this: “Ceftazidime / Avibactam in ventilator-associated pneumonia due to difficult-to-treat non-fermenter Gram-negative bacteria in COVID-19 patients: a case series"

Introduction:

- remove all "trivial" explanations (“Ceftazidime is a third-generation, broad-spectrum cephalosporin that exerts its antibacterial effect by binding to penicillin-binding proteins and inhibiting peptidoglycan crosslinking during cell wall synthesis, leading to bacterial cell lysis and death”), which lengthen the text and are too obvious. If the target audience deals with Ceftazidime-Avibactam, it is necessary to bring the technical level of the entire paper to a higher level, avoiding "trivialities". The management of DTT is of interest to ICU and Infectious Medicine experts, therefore the "obvious" must be removed.

- Technical definition of DTTs should be transferred in M&M section, remaining drier in the description during this part.

- the whole second part of the introduction should be transferred and used for the Discussion.

- In the aim, Authors should enter the words “Case Series”

Material & Methods:

Informed Consent (IC): the sentence concerning the IC is absolutely to be removed, because it is closer to the MK-Ultra project than to the Helsinki declaration. If informed consent has not been requested because it is a case series, because it is a "quality study" or due to daily technical-practical contingencies, I would fail to explain the reasons, describing that "given the descriptive nature of the paper, informed consent has been waived by the IRB" or something like this.

Even if it is a Case Series, it is necessary to insert the statistical part, even if it is short; if instead it is a review, it is a question of redesigning the paper. It is necessary to enter the descriptive statistic, mean/SD or median/IQR according to data distribution (the Kolmorogoff-Smirnov test has been performed? How is it possible to use mean/median? If yes, it must at least be declared in the statistical part of the paper). It’s a very small part to insert, but Authors have to complete it.

Results:

In the first sentence of the Results section, Authors should enter the SOFA score (according to data distribution), the BMI requires a unit of measurement (Kg/m2), etc…

- the pathogens description and the second part must be more concise and described in a more scientific-technical and less discursive way.

Discussion:

- please remove everywhere the world “study”, because this is not a trial.

- the discussion is interesting but too long and takes on a review character; it is necessary that the focus is restricted to fewer topics, which, although interesting are not found in the results (such as the relationship between MIC and pharmacological dose). Authors should be more factual and linked to the results (is it possible to insert the MIC and pharmacokinetic data and analyze them together? in my opinion this is very interesting), or transform the entire paper into a review, but in this case without reporting any patient data.

- Similarly, 80 references are too many for a case series;

As said initially, the paper is valuable in content and revision, but it is too rich and therefore becomes confusing. I suggest that the Authors turn this paper into a short, dry, technical and results-based Case Series, and then take the same valuable information that they have entered here, but which is excessive, and prepare a nice review on the topic, possibly enhancing the literature. At the end of this substantial review, I am available to review the paper.

Author Response

Please, see the attached file with point-by-point responses to your comments, thank you

Reviewer 2 Report

Authors used a case series study to identify the potential effect of Ceftazidine/avibactam in treatment of ventilator associated pneumonia of Gram-negative bacteria among COVID-19 patients. Because the ventilator associated pneumonia may have high case fatality rate, it may be meaningful to analyze the impact of treatment by Ceftazidine/avibactam. However, this article has not been fully answered some of questions due to the insufficient description and in adequate statistical analysis.

First, authors may use simple “mortality rate”, dividing the number of deaths by the total number of patients. However, there may be loss of follow-up including hospital discharge, and these censored cases may affect mortality rate (and case fatality rate). Therefore, authors should add Kaplan-Meier curve as well as days until end of follow-up in table 3.

Second, there is a fluctuation in the description (e.g., fatality rate (L21) vs mortality rate (L34), difficult-to-treat (L3) vs hard-to-treat (L209), real world (L99) vs real life (L104), and b-lactamase (L80) vs β-lactamase (L66)) in this manuscript. Thus, authors should carefully describe manuscript.

Finally, authors did not cite the reference for the description “a dramatic shortage occurred in December 2020 when C/T was withdrawn following a bacterial contamination that occurred during manufacturing process.”, but it may be difficult to understand this shortage for readers in other countries and future readers. Therefore, authors should cite reference for this description.

Minor comments

L84. “…program[11,19-21] . CZA…” may be typo.

L118. “… according to the last the definition …” may be typo.

L128. “…Recurrence of  the…” may be typo.

L145. “…> 24.99…” may be typo of “…>24.99 kg/m2 …”.

Table 1. Abbreviation should be added for PT, ICU and SOFA.

Tittle and so on. COVID19 may be typo of COVID-19.

L487. Informed consent statement and IRB statement should be added.

Author Response

(The authors gave the same response as above.)

Reviewer 3 Report

A significant paper. These studies are currently required to assess whether combination therapy with antibiotics can improve the treatment of patients infected with non-fermenters.

I have only a few corrections to the manuscript:

I propose to include Table 3: patients’ outcomes in the results section.

L70-71:  please provide the reference

L132-133:  All the patients included in our study were treated with CZA; the choice to use a single or combination therapy was based on the patients’ clinical conditions and the microbiological characteristics of the isolates. – this statement should be explained.

L136: a renal adjustment dose according to the manufacturer’s recommendations. Or according to the SPC of the medicinal product?

Author Response

(The authors gave the same response as above.)

Round 2

Reviewer 1 Report

I thank the Editor for once again giving me the opportunity to revise this interesting text by Italian colleagues. I find the work done of revision optimal, even if some minor changes remain to be implemented.

Methods

Page 3: the first time an abbreviation is entered, it must be written out in full. There are abbreviations (GRGNB, SPC, etc.) that have no such explanation at the beginning. Please made a general check and correct all of them.

Statistical section:

If the authors agree, I would write as follows: "Descriptive statistics were performed; continuous variables were presented as number (n), median and interquartile range (IQR), minimum (min) and maximum (max), while categorial variables were presented as frequency/percentage (n/%). A Kaplan-Meier curve was performed concerning 30-days overall survival analysis, starting from the day when Ceftazidime/avibactam was started for treatment. Statistical analysis was performed using STATA(R) software etc."

Results:

Page 4, line 176: replace "clinical cases" with "case series".

Figure 1: the "analysis time" legend has to be changed to "days from starting treatment"; in addition, the values of the remaining patients have to be entered in the abscissas. This is the standard presentation of a Kaplan-Meyer curve.

Discussion:

The length of the paper and the choice of putting together case series & literature review, I personally find misleading and unscientific; this technical judgement is outside the analysis of the content, which was and remains very good and interesting. I still confirm that I would not put the two elements together. However, if the Editor accepts the publication of this article on this 'mixed type' of case series & review as far as the journal's editorial choices are concerned, nothing stands in my way, although I reserve the right to point out this point.

Conclusions:

The conclusions, although very interesting, do not reflect the results. If we are talking about a case series, where epidemiological data are analyzed with descriptive statistics, the focus of the conclusion must be similar. The discourse on CZA in the Conclusion should be limited to one final sentence. The rest of the considerations, I would move elsewhere.

Since References mainly reflect published scientific articles, according to Instruction for Authors all REF must be presented like this:

Journal Articles:
1. Author 1, A.B.; Author 2, C.D. Title of the article. Abbreviated Journal Name Year, Volume, page range.

Author Response

Dear Reviewer, please see the attachment below
